# Convolutional Tensor-Train LSTM for Spatio-Temporal Learning

**Jiahao Su**[*12†]    **Wonmin Byeon**[*2]    **Jean Kossaifi**[2]
jiahaosu@umd.edu, {wbyeon,jkossaifi}@nvidia.com

**Furong Huang**[1]    **Jan Kautz**[2]    **Anima Anandkumar**[2]
furongh@cs.umd.edu, {jkautz,aanandkumar}@nvidia.com

[1]University of Maryland, College Park, MD    [2]NVIDIA Research, Santa Clara, CA

## Abstract

Learning from spatio-temporal data has numerous applications such as human-behavior analysis, object tracking, video compression, and physics simulation. However, existing methods still perform poorly on challenging video tasks such as long-term forecasting. The gap partially is because these kinds of challenging tasks require learning long-term spatio-temporal correlations in the video sequence. We propose a higher-order convolutional LSTM model that can efficiently learn these correlations with a succinct representation of the history. Our model relies on a novel tensor-train module that performs prediction by combining convolutional features across time. To make computation and memory requirements feasible, we develop a novel convolutional tensor-train decomposition of the higher-order model. This decomposition reduces the model complexity by jointly approximating a sequence of convolutional kernels as a low-rank tensor-train factorization. As a result, our model outperforms existing approaches but uses only a fraction of parameters, including the baseline models. Our results achieve state-of-the-art performance in a wide range of applications and datasets, including the multi-steps video prediction on the Moving-MNIST-2 and KTH action datasets as well as early activity recognition on the Something-Something V2 dataset.

## 1   Introduction

While computer vision has achieved remarkable successes, e.g., on image classification, many real-life tasks remain out-of-reach for current deep learning systems, such as prediction from complex spatio-temporal data. Such data naturally arises in a wide range of applications such as autonomous driving, robot control [1], visual perception tasks such as action recognition [2] or object tracking [3], and even weather prediction [4]. This kind of video understanding problems is challenging, since they require learning spatial-temporal representations that capture both content and dynamics simultaneously.

**Learning from (video) sequences.** Most state-of-the-art video models are based on recurrent neural networks (RNNs), typically some variations of *Convolutional LSTM* (ConvLSTM) where spatio-temporal information is encoded explicitly in each cell [4–7]. These RNNs are first-order Markovian models in nature, meaning that the hidden states are updated using information from the previous time step only, resulting in an intrinsic difficulty in capturing long-range temporal correlations.

---

[*]Equal contribution

[†]This work was done while the first author was an intern at NVIDIA.
  Project page: https://sites.google.com/nvidia.com/conv-tt-lstm

**Incorporating higher-order correlations.** For one-dimensional sequence modeling, Soltani and Jiang [8] and Yu et al. [9] proposed higher-order generalizations of RNNs for long-term forecasting problems. Higher-order RNNs explicitly incorporate an extended history of previous states in each update, which requires higher-order tensors to characterize the transition function (instead of a transition matrix as in the first-order RNNs). However, this typically leads to an exponential blow-up in the complexity of the transition function. This problem is compounded when generalizing ConvLSTM to higher-orders and no prior work explores these generalizations.

**Scaling up with tensor methods.** To avoid the exponential blow-up in the complexity of transition function, tensor decompositions [10] have been investigated within higher-order RNNs [9]. Tensor decomposition avoids the exponential growth of model complexity and introduces an information bottleneck that facilitates effective representation learning. This bottleneck restricts how much information can be passed on from one sub-system to another in a learning system [11, 12]. Previously, low-rank tensor factorization has been used to improve various deep network architectures [13–16]. However, its application has not been explored in the context of spatio-temporal LSTMs. The only approach that leveraged tensor factorization for compact higher-order LSTMs [9] considers exclusively sequence forecasting, which does not naturally extend to general spatio-temporal data.

**Generalizing ConvLSTM to higher-orders.** When extending to higher-orders, we aim to design a transition function that can leverage all previous hidden states and satisfies three properties: **(i)** The operations preserve the spatial structure of the hidden states; **(ii)** The receptive field increases with time. In other words, the longer the temporal correlation is captured, the larger the spatial context should be; **(iii)** Finally, space and time complexities grow at most linearly with the number of times steps. Because previous transition functions in higher-order RNNs were designed for one-dimensional sequence, when directly extended to spatio-temporal data, they do not satisfy all three properties. A direct extension fails to preserve the spatial stricture or increases the complexity exponentially.

**Contributions.** In this paper, we propose a higher-order Convolutional LSTM model for complex spatio-temporal data satisfying all three properties. Our model incorporates a long history of states in each update while preserving their spatial structure using convolutional operations. Directly constructing such a model leads to an exponential growth of parameters in both spatial and temporal dimensions. Instead, our model is made computationally tractable via a novel convolutional tensor-train decomposition, which recursively performs a convolutional factorization of the kernels across time. Besides parameter reduction, this factorization introduces an information bottleneck enabling the learning of better representations. As a result, it achieves better results than previous works with only a fraction of parameters.

We empirically demonstrate our model's performance on several challenging tasks, including early activity recognition and video prediction. We report an absolute increase of 8% accuracy over the state-of-the-art [7] for early activity recognition on the Something-Something v2 dataset. Our model outperforms both 3D-CNN and ConvLSTM by a large margin. We also report a new state-of-the-art for multi-step video prediction on both Moving-MNIST-2 and KTH datasets.

## 2    Background: Convolutional LSTM and Higher-order LSTM

In this section, we briefly review *Long Short-Term Memory* (LSTM), and its generalizations *Convolutional LSTM* for sptio-temporal modeling, and *higher-order LSTM* for learning long-term dynamics.

**Long Short-Term Memory (LSTM) [17]** is a first-order Markovian model widely used in 1D sequence learning. At each time step, an LSTM cell updates its states $\{\boldsymbol{h}(t), \boldsymbol{c}(t)\}$ using the immediate previous states $\{\boldsymbol{h}(t-1), \boldsymbol{c}(t-1)\}$ and the current input $\boldsymbol{x}(t)$ as

$$[\boldsymbol{i}(t); \boldsymbol{f}(t); \tilde{\boldsymbol{c}}(t); \boldsymbol{o}(t)] = \sigma(\boldsymbol{W}\boldsymbol{x}(t) + \boldsymbol{K}\boldsymbol{h}(t-1)); \tag{1a}$$

$$\boldsymbol{c}(t) = \boldsymbol{c}(t-1) \circ \boldsymbol{f}(t) + \tilde{\boldsymbol{c}}(t) \circ \boldsymbol{i}(t), \ \boldsymbol{h}(t) = \boldsymbol{o}(t) \circ \sigma(\boldsymbol{c}(t)), \tag{1b}$$

where $\sigma(\cdot)$ denotes a $\mathsf{sigmoid}(\cdot)$ applied to the *input gate $\boldsymbol{i}(t)$*, *forget gate $\boldsymbol{f}(t)$* and *output gate $\boldsymbol{o}(t)$*, and a $\tanh(\cdot)$ applied to the *memory cell $\tilde{\boldsymbol{c}}(t)$* and *cell state $\boldsymbol{c}(t)$*. $\circ$ denotes element-wise product. LSTMs have two major restrictions: **(a)** only 1D-sequences can be modeled, not spatio-temporal data such as videos; **(b)** they are difficult to capture long-term dynamics as first-order models.

**Convolutional LSTM (ConvLSTM) [4, 18]** addresses the limitation **(a)** by extending LSTM to model spatio-temporal structures within each cell, i.e., the states, cell memory, gates and parameters

are all encoded as high-dimensional tensors:

$$[\mathcal{I}(t); \mathcal{F}(t); \tilde{\mathcal{C}}(t); \mathcal{O}(t)] = \sigma(\mathcal{W} * \mathcal{X}(t) + \mathcal{K} * \mathcal{H}(t-1)), \tag{2}$$

where $*$ defines convolution between states and parameters as in convolutional neural networks.

**Higher-order LSTM (HO-LSTM)** is a higher-order Markovian generalization of the basic LSTM, which partially addresses the limitation **(b)** in modeling long-term dynamics. Specifically, HO-LSTM explicitly incorporates more previous states in each update, replacing the first step in LSTM by

$$[\boldsymbol{i}(t); \boldsymbol{f}(t); \tilde{\boldsymbol{c}}(t); \boldsymbol{o}(t)] = \sigma\left(\boldsymbol{W}\boldsymbol{x}(t) + \Phi\left(\boldsymbol{h}(t-1), \cdots, \boldsymbol{h}(t-N)\right)\right), \tag{3}$$

where $\Phi$ combines $N$ previous states $\{\boldsymbol{h}(t-1), \cdots, \boldsymbol{h}(t-N)\}$ and $N$ is the *order* of the HO-LSTM. Two realizations of $\Phi$ have been proposed: a linear function [8] and a polynomial one [9]:

$$\text{Linear:} \quad \Phi\left(\boldsymbol{h}(t-1), \cdots, \boldsymbol{h}(t-N); \boldsymbol{T}(1), \cdots, \boldsymbol{T}(N)\right) = \sum\nolimits_{i=1}^{N} \boldsymbol{T}(i)\boldsymbol{h}(t-i). \tag{4}$$

$$\text{Polynomial:} \quad \Phi\left(\boldsymbol{h}(t-1), \cdots, \boldsymbol{h}(t-N); \mathcal{T}\right) = \langle \mathcal{T}, \ \boldsymbol{h}(t-1) \otimes \cdots \otimes \boldsymbol{h}(t-N)\rangle. \tag{5}$$

While a linear function requires the numbers of parameters and operations growing linearly in $N$, a polynomial function has space/computational complexity exponential in $N$ if implemented naively.

# 3 Methodology: Convolutional Tensor-Train LSTM

Here, we detail the challenges and requirements for designing a higher-order ConvLSTM. We then introduce our model and motivate the design of each module by these requirements.

## 3.1 Extending ConvLSTM to Higher-orders

We can express a general higher-order ConvLSTM by combining several previous states when computing the gates for each step:

$$\left[\mathcal{I}(t); \mathcal{F}(t); \tilde{\boldsymbol{C}}(t); \mathcal{O}(t)\right] = \sigma\left(\mathcal{W} * \mathcal{X}(t) + \Phi\left(\mathcal{H}(t-1), \cdots, \mathcal{H}(t-N)\right)\right). \tag{6}$$

(1) The spatial structure in the hidden states $\mathcal{H}(t)$'s is preserved by the operations in $\Phi$.

(2) The size of the receptive field for $\mathcal{H}(t-i)$ increases with $i$, the time gap from the current step ($i = 1, 2, \cdots, N$). In other words, the longer temporal correlation captured, the larger the considered spatial context should be.

(3) Both space and time complexities grow *at most* linearly with times steps $N$, i.e., $O(N)$.

**Limitations of previous approaches.** While it is possible to construct a function $\Phi$ by extending the linear function in Eq.(4) or the polynomial function in Eq.(5) to the tensor case, none of these extensions satisfy the all three properties. While the polynomial function with tensor-train decomposition [9] meets requirement **(3)**, the operations do not preserve the spatial structures in the hidden states. On the other hand, augmenting the linear function with convolutions leads to a function:

$$\Phi\left(\mathcal{H}(t-1), \cdots, \mathcal{H}(t-N); \mathcal{K}(1), \cdots, \mathcal{K}(N)\right) = \sum\nolimits_{i=1}^{N} \mathcal{K}(i) * \mathcal{H}(t-i) \tag{7}$$

which does not satisfy requirement **(2)** if all $\mathcal{K}(i)$ contain filters of the same size $K$. An immediate remedy is to expand $\mathcal{K}(i)$ such that its filter size $K(i)$ grows linearly in $i$. However, the resulting function requires $O(N^3)$ space/computational complexity, violating the requirement **(3)**.

## 3.2 Designing an Effective and Efficient Higher-order ConvLSTM

In order to satisfy all three requirements **(1)-(3)** introduced above, and enable efficient learning/inference, we propose a novel *convolutional tensor-train decomposition* (CTTD) that leverages a tensor-train structure [19] to jointly express the convolutional kernels $\{\mathcal{K}(1), \cdots, \mathcal{K}(N)\}$ in Eq.(7) as a series of smaller factors $\{\mathcal{G}(1), \cdots, \mathcal{G}(N)\}$ while maintaining their spatial structures.

**Convolutional Tensor-Train module.** Concretely, let $\mathcal{K}(i)$ be the $i$-th kernel in Eq.(7), of size $[K(i) \times K(i) \times C(i) \times C(0)]$, where $K(i) = i[K(1) - 1] + 1$ is the filter size that increases linearly with $i$; $K(1)$ is the initial filter size; $C(i)$ is the number of channels in $\mathcal{H}(t-i)$; and $C(0)$ is the number of channels for the output of the function $\Phi$ (thus $C(0) = 4 \times C_{\text{out}}$, where $C_{\text{out}}$ is the number of channels of the higher-order ConvLSTM). The CTTD factorizes $\mathcal{K}(i)$ using a subset of factors $\{\mathcal{G}(1), \cdots, \mathcal{G}(i)\}$ up to index $i$ such that

$$\mathcal{K}(i)_{:,:,c_i,c_0} \triangleq \mathsf{CTTD}\left(\{\mathcal{G}(j)\}_{j=1}^{i}\right) = \sum_{c_{i-1}=1}^{C(i-1)} \cdots \sum_{c_1=1}^{C(1)} \mathcal{G}(i)_{:,:,c_i,c_{i-1}} * \cdots * \mathcal{G}(2)_{:,:,c_2,c_1} * \mathcal{G}(1)_{:,:,c_1,c_0}, \tag{8}$$

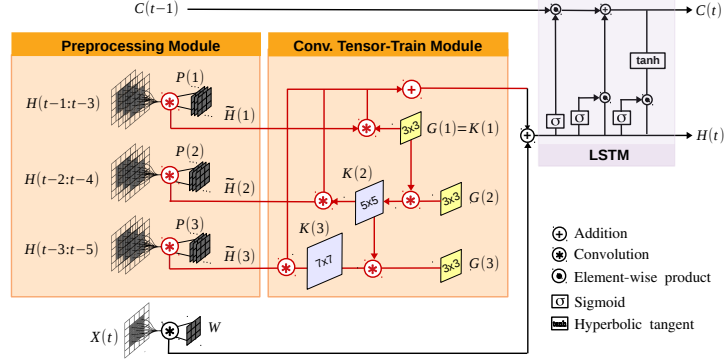

Figure 1: **Convolutional Tensor-Train LSTM**. The *preprocessing module* first groups the previous hidden states into overlapping sets with a sliding window, and reduces the number of channels in each group using a convolutional layer. The *convolutional tensor-train module* takes the results, aggregates their spatio-temporal information, and computes the gates for the LSTM update. The diagram visualizes a Conv-TT-LSTM with one channel. When Conv-TT-LSTM has multiple channels, the addition also accumulates the results from multiple channels.

where $\mathcal{G}(i)$ has size $[K(1) \times K(1) \times C(i) \times C(i-1)]$. The number of factors $N$ is known as the *order of the decomposition*, and the *ranks of the decomposition* $\{C(1), \cdots, C(N-1)\}$ are the channels of the convolutional kernels.

Notice that the same set of factors $\{\mathcal{G}(1), \cdots, \mathcal{G}(N)\}$ is reused to construct all convolutional kernels $\{\mathcal{K}(1), \cdots, \mathcal{K}(N)\}$, such that the number of total parameters grows linearly in $N$. In fact, the convolutional kernel $\mathcal{K}(i+1)$ can be recursively constructed as $\mathcal{K}(i) = \mathcal{G}(i) * \mathcal{K}(i-1)$ with $\mathcal{K}(1) = \mathcal{G}(1)$ and $\mathcal{K}(i)_{:,:,c_i,c_0} = \sum_{c_{i-1}} \mathcal{G}(i)_{:,:,c_i,c_{i-1}} * \mathcal{K}(i-1)_{:,:,c_{i-1},c_0}$ for $i \geq 2$.

This results into in a *convolutional tensor-train module* that we use for function $\Phi$ in Eq.(7):

$$\Phi = \mathsf{CTT}(\mathcal{H}(t-1), \cdots, \mathcal{H}(t-N); \mathcal{G}(1), \cdots, \mathcal{G}(N)) = \sum_{i=1}^{N} \mathsf{CTTD}\big(\{\mathcal{G}(j)\}_{j=1}^{i}\big) * \mathcal{H}(t-i) \quad (9)$$

In Appendix A, we show that the computation of Eq.(9) can be done in linear time $O(N)$, thus the construction of $\mathsf{CTT}$ satisfies all requirements **(1)-(3)**.

**Preprocessing module.** In Eq.(9), we use the raw hidden states $\mathcal{H}(t)$ as inputs to $\mathsf{CTT}$. This design has two limitations: **(a)** The number of past steps in $\mathsf{CTT}$ (i.e., the order of the higher-order ConvLSTM) is equal to the number of factors in CTTD (i.e., the order of the tensor decomposition), which both equal to $N$. It is prohibitive to use a long history, as a large tensor order leads to gradient vanishing/exploding problem in computing Eq.(9); **(b)** All the ranks $C(i)$ are equal to the number of channels in $\mathcal{H}(t)$, which prevents the use of lower-ranks to further reduce the model complexity.

To address both issues, we develop a preprocessing module to reduce the number of steps and channels in previous hidden states before $\mathsf{CTT}$. Suppose the number of steps $M$ is no less than the tensor order $N$ (i.e., $M \geq N$), the preprocessing collects the neighboring steps with a sliding window and reduce it into an intermediate result with $\tilde{C}(i)$ channels:

$$\tilde{\mathcal{H}}(i) = \mathcal{P}(i) * [\mathcal{H}(t-i); \cdots; \mathcal{H}(t-i+N-M)] \quad (10)$$

where $\mathcal{P}(i)$ represents a convolutional layer that maps the concatenation $[\cdot]$ into $\tilde{\mathcal{H}}(i)$.

**Convolutional Tensor-Train LSTM.** By combining all the above modules, we obtain our proposed Conv-TT-LSTM, illustrated in Figure 1 and expressed as:

$$[\mathcal{I}(t); \mathcal{F}(t); \tilde{\boldsymbol{C}}(t); \mathcal{O}(t)] = \sigma\big(\mathcal{W} * \mathcal{X}(t) + \mathsf{CTT}\big(\tilde{\mathcal{H}}(1), \cdots, \tilde{\mathcal{H}}(N); \mathcal{G}(1), \cdots, \mathcal{G}(N)\big)\big) \quad (11)$$

This final implementation has several advantages: it drastically reduces the number of parameters and makes the higher-order ConvLSTM even more compact than the first-order ConvLSTM. The low-rank constraint acts as an implicit regularizer, leading to more generalizable models. Finally, the tensor-train structure inherently encodes the correlations resulting from the natural flow of time [9]. The full procedure can be found in Appendix A (algorithm 2).

# 4 Experiments

Here, we empirically evaluate our approach on several datasets for two different tasks — video prediction and early activity recognition and find out it outperforms existing approaches.

**Evaluation.** For *video prediction*, the model predicts every pixel in the frame. We test our proposed models on the KTH human action dataset [20] with resolution $128 \times 128$ and the Moving-MNIST-2 dataset [2] with resolution $64 \times 64$. All models are trained to predict 10 future frames given 10 input frames and tested to predict $10 - 40$ frames recursively. For *early activity recognition*, we evaluate our approach on the Something-Something V2 dataset. Following [7], we used the subset of 41 categories defined by Goyal et al. [21] (Table 7). The prediction model is trained to predict the next 10 frames given $25\% - 50\%$ of frames, and jointly classify the activity using the learned representations of the prediction model.

**Model architecture.** In all video prediction experiments, we use 12 recurrent layers. For early activity recognition, we follow the framework in [7]. The prediction model consists of two-layers 2D-convolutional encoder and decoder with eight recurrent layers in between. The classifier, which contains two 2D-convolutional layers and one fully-connected layer, takes the last recurrent layer's output and returns a label output. We explain the detailed architecture in Appendix B.

**Loss function.** For video prediction, we optimize an $\ell_1 + \ell_2$ loss $\mathcal{L}_{\text{prediction}} = \|\mathcal{X} - \hat{\mathcal{X}}\|_F^2 + \|\mathcal{X} - \hat{\mathcal{X}}\|_1$, where $\mathcal{X}$ and $\hat{\mathcal{X}}$ are the ground-truth and predicted frames. For early activity recognition, we combine the prediction loss above with an additional cross entropy for classification $\mathcal{L}_{\text{recognition}} = \lambda \cdot \mathcal{L}_{\text{prediction}} + \mathcal{L}_{\text{ce}}(y, \hat{y})$, where $y$ and $\hat{y}$ are the ground-truth and predicted labels. The *weighting factor* $\lambda$ balances the learning representation and exploiting the representation for activity recognition.

**Hyper-parameter selection.** We validate the hyper-parameters of our Conv-TT-LSTM on though a wide grid search on the validation set. Specifically, we consider a base filter size $S = 3, 5$, order of the decomposition $N = 1, 2, 3, 5$, tensor ranks $C(i) = 4, 8, 16$, and number of hidden states $M = 1, 3, 5$. Appendix B contains the details of our hyper-parameter search.

**Efficient Implementation.** Two versions of the implementation are available: the original and the optimized version. In the optimized version, we use multi-threading to accelerate our implementation using the NVIDIA apex library [22]. Furthermore, we adopt fused kernels to speed up the ADAM optimizer [23] and TorchScript to fuse multiplications and additions. Lastly, we use affinity binding to reduce the communication cost between GPUs and CPUs. These modifications speed up training up to four times. Both versions are available online: `https://github.com/NVlabs/conv-tt-lstm`.

## 4.1 Experimental Results

**Multi-frame Video prediction: KTH action dataset.** First, we test our model with human action videos. In Table 2, we report the evaluation on both 20 and 40 frames prediction. Figure 2 (right) shows the model comparisons with SSIM v.s. LPIPS and the model size. **(1)** Our model is consistently better than the ConvLSTM baseline for both 20 and 40 frames prediction. **(2)** While our proposed

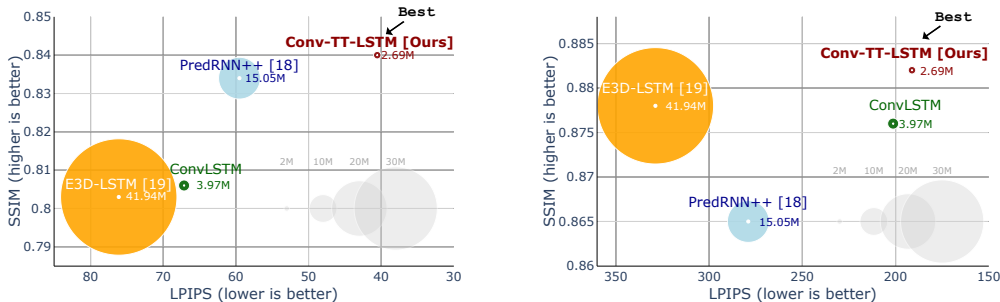

Figure 2: **SSIM v.s. LPIPS scores on Moving MNIST-s2 (left) and KTH action datasets (right). The bubble size is the model size.** The higher SSIM scores and lower LPIPS scores, the better quality of predictions. On both datasets and for both metrics, our approach reaches a significantly better performance than other methods while having only a fraction of the parameters.

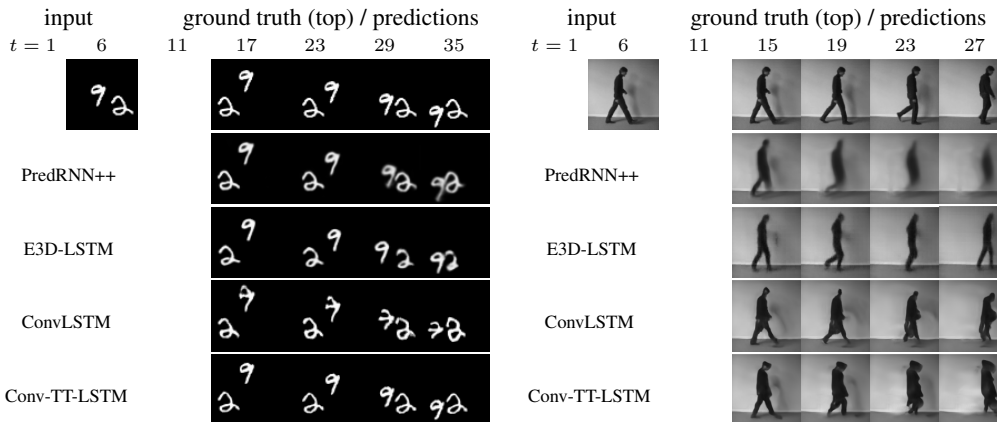

**Figure 3:** **30 frames prediction on Moving-MNIST (left)**, and **20 frame prediction on KTH action datasets (right)** given 10 input frames. The first frames ($t = 1, 11$) are animations. Adobe reader is required to view the animation. Our method generates both semantically plausible and visually crisp images, compared to other approaches.

Conv-TT-LSTMs achieve lower SSIM value compared to the state-of-the-art models in 20 frames prediction, they outperform all previous models in LPIPS for both 20 and 40 frames prediction. Figure 3 (right) shows a visual comparison of our model, ConvLSTM baseline, PredRNN++ [6], and E3D-LSTM [7]. Appendix C includes more examples of visual results. Overall, our model produces sharper frames and better preserves the human silhouettes' details, although there exist slight artifacts over time (shifting). We believe this artifact can be resolved by using a different loss or an additional technique that helps per-pixel motion prediction.

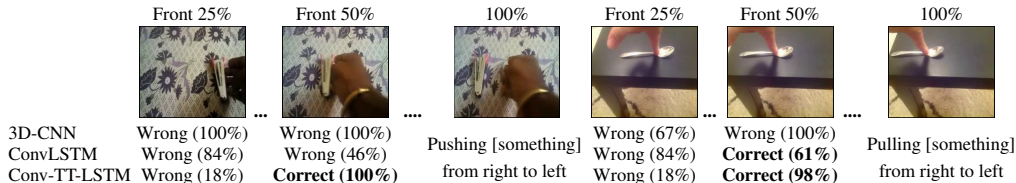

**Figure 4: Examples of Early Activity Recognition Results** given 25% and 50% of frames on the Something-Something V2 dataset, and ($\cdot$) is the confidence for Correct/Wrong prediction.

| Model | Input | Dropping | Holding | MovingLR | MovingRL | Picking | Poking | Pouring | Putting | Showing | Tearing |
|---|---|---|---|---|---|---|---|---|---|---|---|
| 3D-CNN | | 8.5 | 4.7 | 25.8 | 32.6 | 7.5 | 2.9 | 1.9 | 10.3 | 14.0 | 14.5 |
| ConvLSTM | 25% | 8.5 | **7.0** | 27.4 | 38.8 | 16.8 | 5.9 | 1.9 | 12.0 | 7.0 | 21.2 |
| **Conv-TT-LSTM** | | **11.5** | 4.7 | **33.9** | **40.8** | 16.8 | **5.9** | 5.7 | **13.6** | **20.9** | **26.0** |
| 3D-CNN | | 14.6 | 11.6 | 45.2 | 57.1 | 16.8 | 8.8 | 11.3 | 17.4 | 16.3 | 26.0 |
| ConvLSTM | 50% | 21.5 | 7.0 | 43.5 | 47.0 | 15.9 | **14.7** | 5.7 | 20.7 | 16.3 | 30.8 |
| **Conv-TT-LSTM** | | **24.6** | **11.6** | **56.5** | **57.1** | **27.6** | 5.9 | **13.2** | **25.5** | **37.2** | **46.2** |

Table 1: **Per-activity accuracy of early activity recognition on the Something-Something V2 dataset.** We used 41 categories for training. For per-activity evaluation, the 41 categories are grouped into 10 similar activities. The activity mapping are described in [21]. Our model substantially outperforms 3D-CNN and ConvLSTM on long-term dynamics such as Moving or Tearing, while achieves marginal improvement on static activities such as Holding or Pouring.

**Early activity recognition: Something-Something V2 dataset.** To demonstrate that our Conv-TT-LSTM-based prediction model can learn efficient representations from videos, we evaluate the models on early activity recognition on the Something-Something V2 dataset. In this task, a model only observes a small fraction ($25\% - 50\%$) of frames and learns to predict future frames. Based on the learned representations of the beginning frames, the model predicts the full video's overall activity. Intuitively, the learned representation encodes the future information for frame prediction, and the better the quality of the representations, the higher the classification accuracy. As shown in

Table 1 and Table 3 our Conv-TT-LSTM model consistently outperforms the baseline ConvLSTM and 3D-CNN models as well as E3D-LSTM [7] under different ratio of input frames. Our experimental setup and architecture follow [7].

| Method | (10 -> 20) | | | (10 -> 40) | | | Complexities | | |
|---|---|---|---|---|---|---|---|---|---|
| | PSNR | SSIM | LPIPS | PSNR | SSIM | LPIPS | # Params. | # FLOPS | Time(m) |
| *KTH action* | | | | | | | | | |
| ConvLSTM [4] | 23.58 | 0.712 | - | 22.85 | 0.639 | - | 7.58M | 106.6G | - |
| MCNET [24] | 25.95 | 0.804 | - | - | - | - | - | - | - |
| PredRNN++ [6] (retrained [25]) | 28.62 | 0.888 | 228.9 | 26.94 | 0.865 | 279.0 | 15.05M | - | - |
| E3D-LSTM [7] (retrained [26]) | 27.92 | 0.893 | 298.4 | 26.55 | 0.878 | 328.8 | 41.94M | - | - |
| ConvLSTM (baseline) | 28.21 | 0.903 | 137.1 | 26.01 | 0.876 | 201.3 | 3.97M | 55.83G | 28.9 |
| ConvLSTM (classic TTD [27, 28]) | 27.70 | 0.897 | 141.5 | 25.89 | 0.872 | 191.7 | 2.21M | - | - |
| **Conv-TT-LSTM (Ours)** | 28.36 | 0.907 | 133.4 | 26.11 | 0.882 | 191.2 | 2.69M | 37.83G | 74.8 |

| Method | (10 -> 10) | | | (10 -> 30) | | | Complexities | | |
|---|---|---|---|---|---|---|---|---|---|
| | MSE | SSIM | LPIPS | MSE | SSIM | LPIPS | # Params. | # FLOPS | Time(m) |
| *Moving-MNIST* | | | | | | | | | |
| ConvLSTM [4] | 25.22 | 0.713 | - | 38.13 | 0.595 | - | 7.58M | 30.32G | - |
| VPN [29] | 15.65 | 0.870 | - | 31.64 | 0.620 | - | - | - | - |
| PredRNN++ [6] (retrained [25]) | 10.29 | 0.913 | 59.51 | 20.53 | 0.834 | 139.9 | 15.05M | - | - |
| E3D-LSTM [7] (pretrained [26]) | 20.23 | 0.869 | 76.12 | 32.37 | 0.803 | 150.3 | 41.94M | - | - |
| ConvLSTM (baseline) | 18.17 | 0.882 | 67.13 | 33.08 | 0.806 | 140.1 | 3.97M | 15.88G | 14.8 |
| ConvLSTM (classic TTD [27, 28]) | 16.78 | 0.890 | 57.90 | 29.07 | 0.815 | 126.4 | 2.20M | - | - |
| **Conv-TT-LSTM (Ours)** | **12.96** | **0.915** | **40.54** | **25.81** | **0.840** | **90.38** | 2.69M | 10.76G | 29.6 |

Table 2: **Evaluation of multi-steps prediction on the KTH action (top) and Moving-MNIST-2 (bottom) datasets.** Higher PSNR/SSIM and lower MSE/LPIPS values indicate better predictive results. # of FLOPs denotes the multiplications for one-step prediction per sample, and Time(m) represents the clock time (in minutes) required by training the model for one epoch (10,000 samples)

**Multi-frame video prediction: Moving-MNIST-2 dataset.** We also evaluate our model on the Moving-MNIST-2 dataset and show that our model predicts the digits almost correctly in terms of structure and motion (See Figure 3). Table 2 reports the average statistics for 10 and 30 frames prediction, and Figure 2 (left) shows the comparisons of SSIM v.s. LPIPS and the model size. Our Conv-TT-LSTM models **(1)** consistently outperform the ConvLSTM baseline for both 10 and 30 frames prediction *with fewer parameters*; **(2)** outperform previous approaches in terms of SSIM and LPIPS (especially on 30 frames prediction), *with less than one fifth of the model parameters*.

We reproduce the PredRNN++ [6] and E3D-LSTM [7] from the source code [25, 26]. We find that **(1)** PredRNN++ and E3D-LSTM output vague and blurry digits in long-term prediction (especially after 20 steps); **(2)** our Conv-TT-LSTM produces sharp and realistic digits over all steps. An example of visual comparison is shown in Figure 3, and more visualizations can be found in Appendix C.

| Model | Input Ratio | |
|---|---|---|
| | Front 25% | Front 50% |
| 3D-CNN* | 9.11 | 10.30 |
| E3D-LSTM* [7] | 14.59 | 22.73 |
| 3D-CNN | 13.26 | 20.72 |
| ConvLSTM | 15.46 | 21.97 |
| Conv-TT-LSTM (ours) | **19.53** | **30.05** |

| | MSE($\times 10^{-3}$) | SSIM | LPIPS |
|---|---|---|---|
| CTTD with $1 \times 1$ filters (similar to standard TTD) | | | |
| single order | 31.52 | 0.810 | 148.7 |
| order 3 | 34.84 | 0.800 | 151.2 |
| CTTD with $5 \times 5$ filters | | | |
| single order | 33.08 | 0.806 | 140.1 |
| order 3 | **28.88** | **0.831** | **104.1** |

Table 3: **Early activity recognition on the Something-Something V2 dataset** using 41 categories as [7]. (*) indicates the result by [7].

Table 4: **Ablation studies of higher-order Conv-TT-LSTM on Moving-MNIST-2 dataset**. The models are tested for 10 to 30 frames prediction.

# 5 Discussion

In this section, we further justify the importance of the proposed modules, *convolutional tensor-train decomposition* (CTTD) and the *preprocessing module*. We also explain the computational complexity of our model and the difficulties of spatio-temporal learning with Transformer [30].

**Importance of encoding higher-order correlations in a convolutional manner.** Two key differences between CTTD and existing low-rank decompositions are *higher-order decomposition* and *convolutional operations*. To verify their impact, we compare the performance of two ablated models against our CTTD-base model in Table 4. The single order means that the higher-order model is replaced with a first-order model (tensor order = 1). By replacing $5 \times 5$ filters to $1 \times 1$, the convolutions are removed, and the CTTD reduces to a standard tensor-train decomposition. The results show a decrease in performance: the ablated models achieve similar performances of ConvLSTM baseline at best, demonstrating that both higher-order models and convolutional operations are necessary.

**Importance of the preprocessing module.** There could be other ways to incorporate previous hidden states into the CTT module. One is to reduce the number of channels while keeping the number of steps; the other is to reuse all previous states' concatenation for each input to CTT. The former fails due to the gradient vanishing/exploding problem, while the latter has a tube-shaped receptive field that fails to distinguish more recent steps and the ones from the remote history.

**Computational complexity.** Table 2 provides the number of FLOPS for all models. Our Conv-TT-LSTM model has lower computational complexity and fewer parameters than other models under comparison. This efficiency is made possible by a linear algorithm for the convolutional tensor-train module in Eq.(9), which is derived in Appendix A.

**Trade-off between FLOPs and latency.** Notice that a lower FLOPS does not necessarily lead to faster computation since the convolutional tensor-train module is naturally sequential. In Appendix A, we introduce two algorithms. While algorithm 2 significantly decreases the complexity in FLOPs, it also lowers the degree of parallelism. However, algorithm 1 shows how our model can be parallelized. Ideally, these two algorithms can be combined using CUDA multi-streams (execute multiple kernels in parallel): use algorithm 1 for the beginning iterations of $i$ and algorithm 2 for the later ones (the beginning ones have smaller kernel sizes). In our current implementation, we use algorithm 2 to reduce the GPU memory requirement. As stated in Table 2, the run-time is 74.8 minutes for Conv-TT-LSTM (37.83 GFLOPs) vs. 28.9 mins for ConvLSTM (55.83 GFLOPs) per epoch on KTH without any GPU optimization. With an efficient implementation as mentioned in section 4, we manage to decrease them to 27.3 mins (37.83 GFLOPs) for Conv-TT-LSTM vs. 26.2 mins (55.83 GFLOPs) for ConvLSTM. Finally, Conv-TT-LSTM is only slightly slower than ConvLSTM despite the lack of parallelism.

**Classic Tensor-Train Decomposition for RNN compression.** Classic *tensor-train decomposition* (TTD) [19] is traditionally used to compress fully-connected and convolutional layer in a feed-forward network [31, 28], where the parameters in each layer are reshaped into a higher-order tensor and stored in a factorized form. Yang et al. [27] applies this idea to RNNs and compress the input-hidden transition matrix [31]. There are three major differences between our work and Yang et al. [27]:

- Single-order LSTM v.s. Higher-order ConvLSTM. Yang et al. [27] consider a first-order fully-connected LSTM [17] for compression, while our method aims to compress a higher-order convolutional LSTM model.
- Classic decomposition v.s. Convolutional decomposition. Yang et al. [27] relies on the classic TTD, while our proposed *convolutional tensor-train decomposition* (CTTD) factorizes the tensor with convolutions in addition to inner products; our decomposition is designed to preserve spatial structures in spatio-temporal data.
- Compression of input-hidden matrix v.s. hidden-to-hidden convolutional kernels. Yang et al. [27] only compresses input-hidden transition $\boldsymbol{W}$ in LSTM, but our CTTD compresses a sequence of convolutional kernels $\{\mathcal{K}(1), \cdots, \mathcal{K}(N)\}$ for different time steps simultaneously (see Eq.(7)).

To understand the necessity of our design for long-term spatio-temporal dynamics, we develop a compressed ConvLSTM following the same idea in [31, 28, 27], which stores the parameters for input-hidden transition $\mathcal{W}$ in a tensor-train format $\mathcal{W} = \mathsf{TT}(\{\mathcal{G}(i)\}_{i=0}^{N-1})$ (where $N$ denotes the order of the decomposition, i.e., the number of factors):

$$[\mathcal{I}(t); \mathcal{F}(t); \tilde{\mathcal{C}}(t); \mathcal{O}(t)] = \sigma(\mathsf{TT}(\{\mathcal{G}(i)\}_{i=0}^{N-1}) * \mathcal{X}(t) + \mathcal{K} * \mathcal{H}(t-1)) \qquad (12)$$

Since the transition in ConvLSTM is characterized as a convolutional layer, we follow the approach by Garipov et al. [28] and represent $\mathcal{W}$ with size $[K \times K \times C_{\text{out}} \times C_{\text{in}}]$ using $N$ factors: **(1)** The 4-th order tensor $W$ is reshaped to an $2M$-th order tensor $\widetilde{\mathcal{W}}$ with size $[K \times K \times T_1 \cdots \times T_{N-1} \times S_1 \cdots \times S_{N-1}]$ and $C_{\text{out}} = \prod_{i=1}^{N-1} T_i$, $C_{\text{in}} = \prod_{i=1}^{N-1} S_i$; **(2)** The tensor $\widetilde{\mathcal{W}}$ is factorized using TTD as

$$\widetilde{\mathcal{W}}_{i,j,t_1,\cdots,t_{N-1},s_1,\cdots,s_{N-1}} \triangleq \sum_{r_0,\cdots,r_{N-1}} \mathcal{G}(0)_{i,j,r_0} \mathcal{G}(1)_{t_1,s_1,r_0,r_1} \cdots \mathcal{G}(N-1)_{i,j,r_{N-1}} \tag{13}$$

where $\mathcal{G}(0)$ has size $[K \times K \times R_0]$, $\mathcal{G}(i)$ has $[T_i \times S_i \times R_{i-1} \times R_i]$ for $0 < i < N-1$, and $\mathcal{G}(N-1)$ has $[T_{N-1} \times S_{N-1} \times R_{N-1}]$. A comparison against the uncompressed ConvLSTM and our Conv-TT-LSTM is presented in Table 2. We observe that our model outperforms this method on MNIST and KTH (except LPIPS on KTH) with similar number of parameters.

**Transformer for spatio-temporal learning.** Transformer [30] is a popular predictive model based on the attention mechanism, which is very successful in natural language processing [32]. However, the Transformer has prohibitive limitations on video understanding, due to excessive needs for both memory and computation. While language modeling only involves temporal attention, video understanding requires attention to spatial dimensions as well [33]. Moreover, since the attention mechanism does not preserve the spatial structures by design, Transformer additionally requires auxiliary components including an autoregressive module and multi-resolution upscaling when applied on spatial data [34, 35, 33]. Our Conv-TT-LSTM incorporates a broad spatio-temporal context, but with a compact, efficient and structure-preserving operator without additional components.

## 6   Related Work

**Tensor decompositions.** Tensor decompositions such as CP, Tucker or Tensor-Train [36, 19], are widely used for dimensionality reduction [37] and learning probabilistic models [10]. These tensor factorization techniques have also been widely used in deep learning to improve performance, speed-up computation, and compress the deep neural networks [13, 14, 31, 38–41, 16], recurrent networks [42, 27] and Transformers [43]. Yang et al. [27] has proposed tensor-train RNNs to compress both inputs-states and states-states matrices within each cell with TTD by reshaping the matrices into tensors, and showed improvement for video classification.

Departing from prior works that rely on existing, well-established tensor decompositions, here we propose a novel *convolutional tensor-train decomposition* (CTTD) designed to enable efficient and compact higher-order convolutional recurrent networks. Unlike Yang et al. [27], we aim to compress higher-order ConvLSTM, rather than first-order fully-connected LSTM. We further propose Convolutional Tensor-Train decomposition to preserve spatial structure after compression.

**Spatio-temporal prediction models.** Prior prediction models have focused on predicting short-term video [44, 45] or decomposing motion and contents [46, 24, 47, 48]. Many of these works use ConvLSTM as a base module, which deploys 2D convolutional operations in LSTM to efficiently exploit spatio-temporal information. Some works modified the standard ConvLSTM to better capture spatio-temporal correlations [5, 6]. Byeon et al. [45] demonstrated strong performance using a deep ConvLSTM network as a baseline, and we adopt this base architecture in the present paper.

## 7   Conclusion

In this paper, we proposed a fully-convolutional higher-order LSTM model for spatio-temporal data. To make the approach computationally and memory feasible, we proposed a novel convolutional tensor-train decomposition that jointly parameterizes the convolutions and naturally encodes temporal dependencies. The result is a compact model that outperforms prior work on video prediction, including something-something V2, moving-MNIST-2, and KTH action datasets. Going forward, we plan to investigate our CTT module in a framework that spans not only higher-order RNNs but also Transformer-like architectures for efficient spatio-temporal learning.

## Funding Disclosure

This work was done while the author, Jiahao Su, was an intern at NVIDIA. Su was also partially supported by the startup fund from Department of Computer Science of University of Maryland and National Science Foundation IIS-1850220 CRII Award 030742- 00001. The author, Furong Huang, was supported by Adobe, Capital One, and JP Morgan faculty fellowships.

## Impact Statement

In this paper, the authors introduce Convolutional Tensor-Train LSTM model for spatio-temporal learning. Our model can be applied to any spatio-temporal data, e.g., physical system simulation and video understanding.

For physical system simulation, our model could be used in weather or turbulence prediction, where no simple physics rule can be used to anticipate the future. The potential in these applications could reduce loss by extreme weathers, and the chance of aircraft encountering violent turbulence. For video understanding, our model can be applied to a wide range of applications including autonomous driving, robot control, human behavior analysis and object tracking.

While these applications greatly relieve humans from tedious and repeat laboring, they have raised questions in the society. For example, (1) faulty predictions in autonomous driving systems - do we have safeguards in place? (2) human tracking and behavior analysis - are we protecting privacy? (3) finally, object tracking - how are we regulating? Therefore, it is crucial to consider whether the technology could be misused before they are deployed, and what needs to be in place to avoid an undesired consequence.

We suggest the researcher in physical sciences and social sciences to investigate questions such as:

- Can a machine learning approach simulate a physical system given sufficient data? If not, to what extent the physical system can be learned?
- How to systematically verify the capacity of a machine learning model, such that certain behavior can be prohibited before deployment?
- How to define the responsibility if an autonomous system produces an undesired outcome (for example, car crash and personal information leakage)?

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
