[Supplementary Material]

# Appendix: Convolutional Tensor-Train LSTM for Spatio-temporal Learning

Jiahao Su[*12†]        Wonmin Byeon[*2]        Jean Kossaifi[2]

Furong Huang[1]        Jan Kautz[2]        Anima Anandkumar[2]

jiahaosu@umd.edu, {wbyeon,jkossaifi}@nvidia.com

furongh@cs.umd.edu, {jkautz,aanandkumar}@nvidia.com

[1]University of Maryland, College Park, MD    [2]NVIDIA Research, Santa Clara, CA

[*]Equal contribution  [†]This work was done while the first author was an intern at NVIDIA.

Project page: https://sites.google.com/nvidia.com/conv-tt-lstm

In the supplementary material, we first provide a constructive proof that our approach can be computed with linear complexity in time. We then provide complete implementation details for all experiments and perform additional ablation studies of our model, demonstrating that our Conv-TT-LSTM model is general and outperforms regular ConvLSTM with varying settings. Finally, we provide additional visualizations of our experimental results.

To facilitate the reading of our paper, we provide a Table of notations in Table 5.

| Symbol | Meaning | Value or Size |
|---|---|---|
| $H$ | Height of feature map | |
| $W$ | Width of feature map | |
| $C_{\text{in}}$ | # of input channels | - |
| $C_{\text{out}}$ | # of output channels | |
| $t$ | Current time step | - |
| $\mathcal{W}$ | Weights for $\mathcal{X}(t)$ | $[K \times K \times 4C_{\text{out}} \times C_{\text{in}}]$ |
| $\mathcal{X}(t)$ | Input features | $[H \times W \times C_{\text{in}}]$ |
| $\mathcal{H}(t)$ | Hidden state | |
| $\mathcal{C}(t)$ | Cell state | |
| $\mathcal{I}(t)$ | Input gate | |
| $\mathcal{F}(t)$ | Forget gate | $[H \times W \times C_{\text{out}}]$ |
| $\tilde{\mathcal{C}}(t)$ | Cell memory | |
| $\mathcal{O}(t)$ | Output gate | |
| $\Phi$ | Mapping function for higher-order RNN | - |
| $M$ | order of higher-order RNN | |
| $N$ | Order of CTTD | $M \geq N$ |
| $K$ | Initial filter size | |
| $K(i)$ | Filter size in $\tilde{\mathcal{K}}(i)$ | $K(0) = K$ |
| $C(i)$ | # channels in $\tilde{\mathcal{H}}(i)$ | $C(0) = 4C_{\text{out}}$ |
| $\mathcal{G}(i)$ | Factors in the CTTD | $[K(0) \times K(0) \times C(i) \times C(i-1)]$ |
| $D$ | Size of sliding window | $D = M - N + 1$ |
| $\mathcal{P}(i)$ | Preprocessing kernel | $[D \times K \times K \times C_{\text{out}} \times C(i)]$ |
| $\tilde{\mathcal{H}}(i)$ | Pre-processed hidden state | $[H \times W \times C(i)]$ |
| $\mathcal{K}(i)$ | Weights for $\tilde{\mathcal{H}}(i)$ | $[K(i) \times K(i) \times C(i) \times C(0)]$ |

Table 5: **Table of notations.**

# A An Efficient Algorithm for Convolutional Tensor-Train Module

This section proves that our convolutional tensor-train module, CTT (Eq.(9) in main paper), can be evaluated with linear computational complexity. Our proof is constructive and readily provides an algorithm for computing CTT in linear time.

First, let's recall the formulation of the CTT function:

$$\Phi = \text{CTT}(\mathcal{H}(t-1), \cdots, \mathcal{H}(t-N); \mathcal{G}(1), \cdots, \mathcal{G}(N)) = \sum_{i=1}^{N} \mathcal{K}(i) * \mathcal{H}(t-i), \tag{14}$$

where $\mathcal{K}(i)$ is the $i$-th kernel with size $[K(i) \times K(i) \times C(i) \times C(0)]$ ($K(i) = i[K(1) - 1] + 1$ is the filter size that increases linearly with $i$; $K(1)$ is the initial filter size; $C(i)$ is the number of channels in $\mathcal{H}(t-i)$; and $C(0)$ is the number of channels for the output of the function $\Phi$). Moreover, each kernel $\mathcal{K}(i)$ is factorized by *convolutional tensor-train decomposition* (CTTD).

$$\mathcal{K}(i)_{:,:,c_i,c_0} \triangleq \text{CTTD}\left(\{\mathcal{G}(j)\}_{j=1}^{i}\right) = \sum_{c_{i-1}=1}^{C(i-1)} \cdots \sum_{c_1=1}^{C(1)} \mathcal{G}(i)_{:,:,c_i,c_{i-1}} * \cdots * \mathcal{G}(1)_{:,:,c_1,c_0}, \ \forall i \in [N]. \tag{15}$$

However, a naive algorithm that first reconstruct all the kernels $\mathcal{K}(i)$, then applies Eq.(14) results in computational complexity of $O(N^3)$, as illustrated in Eq.(1). To scale our approach to higher-order models (i.e., larger $N$), we need a more efficient implementation of the function CTT.

---

**Algorithm 1: Convolutional Tensor-Train LSTM** (Original: $T(N) = O(N^3)$).

---

**Input:** current input $\mathcal{X}(t)$, previous cell state $\mathcal{C}(t-1)$,
  $M$ previous hidden states $\{\mathcal{H}(t-1), \cdots, \mathcal{H}(t-M)\}$
**Output:** new hidden state $\mathcal{H}(t)$, new cell state $\mathcal{C}(t)$
**Initialization:** $\mathcal{K}(0) = 1$; $\mathcal{V} = 0$
```
/* Convolutional Tensor-Train (CTT) module                              */
```
**for** $i = 1$ **to** $N$ **do**
  ```
  /* preprocessing module                                            */
  // compress the states from a sliding window
  ```
  $\tilde{\mathcal{H}}(i) = \mathcal{P}(i) * [\mathcal{H}(t-i); \cdots; \mathcal{H}(t-i+N-M)]$
  ```
  // recursively construct the kernel
  ```
  $\mathcal{K}(i) = \mathcal{G}(i) * \mathcal{K}(i-1)$
  ```
  // accumulate the output
  ```
  $\mathcal{V} = \mathcal{V} + \mathcal{K}(i) * \tilde{\mathcal{H}}(i)$
**end for**
```
/* Long-Short Term Memory (LSTM)                                        */
```
$\left[\mathcal{I}(t); \mathcal{F}(t); \tilde{\mathcal{C}}(t); \mathcal{O}(t)\right] = \sigma(\mathcal{W} * \mathcal{X}(t) + \mathcal{V})$
$\mathcal{C}(t) = \mathcal{C}(t-1) + \tilde{\mathcal{C}}(t) \circ \mathcal{I}(t)$; $\mathcal{H}(t) = \mathcal{O}(t) \circ \sigma(\mathcal{C}(t))$
return $\mathcal{H}(t), \mathcal{C}(t)$

---

**Recursive evaluation.** We will prove that CTT can be evaluated backward recursively using

$$\mathcal{V}(i-1)_{:,:,c_{i-1}} = \sum_{c_i=1}^{C(i)} \mathcal{G}(i)_{:,:,c_i,c_{i-1}} * \left(\mathcal{V}(i)_{:,:,c_i} + \mathcal{H}(i)_{:,:,c_i}\right), \ i = N, N-1, \cdots, 0 \tag{16}$$

where $\mathcal{V}(N)$ is initialized as zeros, and the final output of CTT is equal to $\mathcal{V}(0)$.

*Proof.* First, we note that $\mathcal{K}(i)$ can be represented recursively in terms of $\mathcal{K}(i-1)$ and $\mathcal{G}(i)$:

$$\mathcal{K}(i)_{:,:,c_i,c_0} = \sum_{c_{i-1}=1}^{C(i-1)} \mathcal{G}(i)_{:,:,c_i,c_{i-1}} * \mathcal{K}(i-1)_{:,:,c_{i-1},c_0} \tag{17}$$

with $\mathcal{K}(1) = \mathcal{G}(1)$. Next, we aim to **inductively** prove the following holds for any $n \in [N]$:

$$\Phi_{:,:,c_0} = \sum_{i=1}^{n} \sum_{c_i=1}^{C(i)} \mathcal{K}(i)_{:,:,c_i,c_0} * \mathcal{H}(t-i)_{:,:,c_i} + \sum_{c_n=1}^{C(n)} \mathcal{K}(n)_{:,:,c_n,c_0} * \mathcal{V}(n)_{:,:,c_n}, \qquad (18)$$

and therefore it holds for $n = 1$, $\Phi_{:,:,c_0} = \sum_{c_1=1}^{C(1)} \mathcal{G}(1)_{:,:,c_1,c_0} * (\mathcal{V}(1)_{:,:,c_1} + \mathcal{H}(1)_{:,:,c_1}) = \mathcal{V}(0)_{:,:,c_0}$.

Notice that the case $n = N$ is obvious by the definition of CTT and the zero initialization of $\mathcal{V}(N)$. Therefore the remaining of this proof is to induce the case $n = N - 1$ from $n = N$.

$$\Phi_{:,:,c_0} = \sum_{i=1}^{N} \sum_{c_i=1}^{C(i)} \mathcal{K}(i)_{:,:,c_i,c_0} * \mathcal{H}(t-i)_{:,:,c_i} + \sum_{c_N=1}^{C(N)} \mathcal{K}(N)_{:,:,c_N,c_0} * \mathcal{V}(N)_{:,:,c_N} \qquad (19)$$

$$= \sum_{i=1}^{N-1} \sum_{c_i=1}^{C(i)} \mathcal{K}(i)_{:,:,c_i,c_0} * \mathcal{H}(t-i)_{:,:,c_i} + \underbrace{\sum_{c_N=1}^{C(N)} \mathcal{K}(N)_{:,:,c_N,c_0} * (\mathcal{H}(N)_{:,:,c_N} + \mathcal{H}(N)_{:,:,c_N})} \qquad (20)$$

Notice that the second term can be rearranged as

$$\sum_{c_N=1}^{C(N)} \mathcal{K}(N)_{:,:,c_N,c_0} * (\mathcal{H}(N)_{:,:,c_N} + \mathcal{H}(N)_{:,:,c_N}) \qquad (21)$$

$$= \sum_{c_N=1}^{C(N)} \underbrace{\left[ \sum_{c_{N-1}=1}^{C(N-1)} \mathcal{G}(N-1)_{:,:,c_N,c_{N-1}} * \mathcal{K}(N-1)_{:,:,c_{N-1},c_0} \right]}_{\mathcal{K}(N)_{:,:,c_N,c_0}, \text{ by Eq.(17)}} * (\mathcal{H}(N)_{:,:,c_N} + \mathcal{H}(N)_{:,:,c_N}) \qquad (22)$$

$$= \sum_{c_{N-1}=1}^{C_{N-1}} \mathcal{K}(N-1)_{:,:,c_{N-1},c_0} * \underbrace{\left[ \sum_{c_N=1}^{C(N)} \mathcal{G}(N-1)_{:,:,c_N,c_{N-1}} * (\mathcal{H}(N)_{:,:,c_N} + \mathcal{H}(N)_{:,:,c_N}) \right]}_{\mathcal{V}(N-1)_{:,:,N-1}, \text{ by Eq.(16)}} \qquad (23)$$

$$= \sum_{c_{N-1}=1}^{C(N-1)} \mathcal{K}(N-1)_{:,:,c_{N-1},c_0} * \mathcal{V}(N-1)_{:,:,N-1} \qquad (24)$$

where Eq.(22) uses the recursive formula in Eq.(17), and Eq.(23) is by definition of $\mathcal{V}(N-1)$ in Eq.(16). Therefore, we show that the case $n = N - 1$ also holds

$$\Phi_{:,:,c_0} = \sum_{i=1}^{N-1} \sum_{c_i=1}^{C(i)} \mathcal{K}(i)_{:,:,c_i,c_0} * \mathcal{H}(t-i)_{:,:,c_i} + \sum_{c_{N-1}=1}^{C(N-1)} \mathcal{K}(N-1)_{:,:,c_{N-1},c_0} * \mathcal{V}(N-1)_{:,:,N-1} \qquad (25)$$

which completes the induction from $n = N$ to $n = N - 1$. $\qquad\square$

# B   Experimental Details

This section provides detailed setups of all experiments (datasets, model architectures, learning strategies, and evaluation metrics) for video prediction and early activity recognition.

## B.1   Preprocessing Module

In the main paper, we use a sliding window to concatenate consecutive states in the preprocessing module (section 3). In the discussion (section 5), we argued that other possible approaches are less effective in preserving spatio-temporal structure than our *sliding window approach*. Here, we discuss an alternative approach previously proposed for non-convolutional higher-order RNN [9], which we name as *fixed window approach*. We will compare these two approaches in computational complexity, temporal structure-preserving, and predictive performance.

**Algorithm 2: Convolutional Tensor-Train LSTM** (Accelerated: $T(N) = O(N)$).

---

**Input:** current input $\mathcal{X}(t)$, previous cell state $\mathcal{C}(t-1)$,
$\qquad$ $M$ previous hidden states $\{\mathcal{H}(t-1), \cdots, \mathcal{H}(t-M)\}$
**Output:** new hidden state $\mathcal{H}(t)$, new cell state $\mathcal{C}(t)$
**Initialization:** $\mathcal{K}(0) = 1$; $\ \mathcal{V}(N) = 0$

```
/* Convolutional Tensor-Train (CTT) module                            */
```
**for** $i = N$ **to** $1$ **do**
$\quad$
```
    /* preprocessing module                                           */
    // compress the states from a sliding window
```
$\quad$ $\tilde{\mathcal{H}}(i) = \mathcal{P}(i) * [\mathcal{H}(t-i); \cdots; \mathcal{H}(t-i+N-M)]$
$\quad$
```
    // recursively compute the intermediate results
```
$\quad$ $\mathcal{V}(i-1) = \mathcal{G}(i) * \left( \mathcal{V}(i) + \tilde{\mathcal{H}}(i) \right);$
**end for**
```
/* Long-Short Term Memory (LSTM)                                       */
```
$\left[ \mathcal{I}(t); \mathcal{F}(t); \tilde{\boldsymbol{C}}(t); \mathcal{O}(t) \right] = \sigma \left( \mathcal{W} * \mathcal{X}(t) + \mathcal{V}(0) \right)$
$\mathcal{C}(t) = \mathcal{C}(t-1) + \tilde{\mathcal{C}}(t) \circ \mathcal{I}(t); \ \mathcal{H}(t) = \mathcal{O}(t) \circ \sigma(\mathcal{C}(t))$
return $\mathcal{H}(t), \mathcal{C}(t)$

---

**Fixed window approach.** With fixed window approach, $M$ previous steps $\{\mathcal{H}(t-1), \cdots, \mathcal{H}(t-M)\}$ are first concatenated into a single tensor, which is then repeatedly mapped to $N$ inputs $\{\tilde{\mathcal{H}}(1), \cdots, \tilde{\mathcal{H}}(N)\}$ to the CTT module.

$$\textbf{Fixed Window (FW):} \quad \tilde{\mathcal{H}}(i) = \mathcal{P}(i) * [\mathcal{H}(t-1); \cdots; \mathcal{H}(t-N)] \qquad (26\text{a})$$

$$\textbf{Sliding Window (SW):} \quad \tilde{\mathcal{H}}(i) = \mathcal{P}(i) * [\mathcal{H}(t-i); \cdots; \mathcal{H}(t-i+N-M)] \qquad (26\text{b})$$

For comparison, we list both equations for the fixed window approach and the sliding window approach. We also illustrate these two approaches in Figure 5.

**Drawbacks of fixed window approach. (a)** The fixed window approach has a larger window size than the sliding window approach, thus requires more parameters in the preprocessing kernels and higher computational complexity. **(b)** More importantly, the fixed window approach does not preserve the chronological order of the preprocessed states; unlike sliding window approach, the index $i$ for $\tilde{\mathcal{H}}(i)$ in fixed window approach cannot reflect the time for the compressed states. Actually, all preprocessed states $\tilde{\mathcal{H}}(1), \cdots, \tilde{\mathcal{H}}(M)$ are equivalent, which violates the property (2) in designing our convolutional tensor-train module (Section 3.1). **(c)** In Table 8, we compare these two approaches on Moving-MNIST-2 under the same experimental setting, and we find that the sliding window approach performs slightly better than the fixed window. We choose sliding window approach in our implementation of the preprocessing module for all the reasons above.

## B.2 Model Architectures

**Video prediction.** All experiments use a 12-layers ConvLSTM / Conv-TT-LSTM with 32 channels for the first and last 3 layers, and 48 channels for the 6 layers in the middle. A convolutional layer is applied on top of all recurrent layers to compute the predicted frames, followed by an extra sigmoid layer for the KTH action dataset. Following Byeon et al. [45], we add two skip connections performing concatenation over channels between (3, 9) and (6, 12) layers. An illustration of the network architecture is included in Figure 6a. All convolutional kernels are initialized by Xavier's normalized initializer [49] and initial hidden/cell states are initialized as zeros.

**Early activity recognition.** Following [7], the network architecture consists of four modules: a 2D-CNN encoder, a video prediction network, a 2D-CNN decoder and a 3D-CNN classifier, as illustrated in Figure 6b. **(1)** The 2D-CNN encoder has two 2-strided 2D-convolutional layers with 64 channels, which reduce the resolution from $224 \times 224$ to $56 \times 56$, and **(2)** the 2D-CNN decoder contains two 2-strided transposed 2D-convolutional layers with 64 channels, which restore the resolution from

(a) Sliding window approach (final implementation)  (b) Fixed window approach (alternative)

Figure 5: **Variations of proprocessing modules.**

$56 \times 56$ to $224 \times 224$. **(3)** The video prediction network is miniature version of Figure 6a, where the number of layers in each block is reduced to $2$. In the experiments, we evaluate three realizations of each layer: ConvLSTM, Conv-TT-LSTM or causal 3D-convolutional layer. **(4)** The 3D-CNN classifier takes the last $16$ frames from the input and predicts a label for the $41$ categories. The classifier contains two 2-strided 3D-convolutional layers with stride $2$ and $128$ channels, each of which is followed by a 3D-pooling layer. These layers reduce the resolution from $56 \times 56$ to $7 \times 7$, and the output feature is fed into a two-layer perceptron with $512$ hidden units for a predictive label.

(a) **Prediction model**  (b) **Recognition model**

Figure 6: **Network architecture for video prediction and early activity recognition tasks.**

### B.3 Training Strategy

To facilitate training, we argue for a careful choice of learning scheduling and gradient clipping. Specifically, various *learning scheduling techniques*, including learning rate decay, scheduled sampling [50], and curriculum learning with varying weighting factors, are added during training. **(1)** For *video prediction*, we use learning rate decay along with scheduled sampling, where scheduled sampling starts if the model does not improve for a few epochs in terms of validation loss. **(2)** For *early activity recognition*, we combine learning rate decay with weighting factor decay, where the weighting factor decreases linearly $\lambda := \max(\lambda - \epsilon, 0)$ on the plateau. **(3)** We also found *gradient clipping* essential for higher-order models. We train All models with ADAM optimizer [23]. In the initial experiments, we found that our models are unstable at a high learning rate $1e^{-3}$, but learn poorly at a low learning rate of $1e^{-4}$. Consequently, we use gradient clipping with learning rate of $1e^{-3}$, with a clipping value of $1$ for all experiments.

## B.4 Hyper-parameters Selection

Table 6 summarizes our search values for different hyper-parameters for Conv-TT-LSTM. **(1)** For filter size $K$, we found models with larger filter size $K = 5$ consistently outperform the ones with $K = 3$. **(2)** For learning rate, we found that our models are unstable at a high learning rate such as $10^{-3}$, but learn poorly at a low learning rate $10^{-4}$. Consequently, we use gradient clipping with learning rate $10^{-3}$, with clipping value 1 for all experiments. **(3)** While the performance typically increases as the order grows, the model suffers gradient instability in training with a high order, e.g., $N = 5$. Therefore, we choose the order $N = 3$ for all Conv-TT-LSTM models. **(4)(5)** For small ranks $C(i)$ and steps $M$, the performance increases monotonically with $C(i)$ and $M$. But the performance stays on plateau when we further increase them, therefore we settle down at $C(i) = 8, \forall i$ and $M = 5$ for all experiments.

| Filter size $K$ | Learning rate | Order of CTTD $N$ | Ranks of CTTD $\mathcal{C}(i)$ | Time steps $M$ |
|---|---|---|---|---|
| $\{3, 5\}$ | $\{10^{-4}, 5 \times 10^{-4}, 10^{-3}\}$ | $\{1, 2, 3, 5\}$ | $\{4, 8, 16\}$ | $\{1, 3, 5\}$ |

Table 6: Hyper-parameters search values for Conv-TT-LSTM experiments.

Similarly, Table 7 summarize the hyper-parameters search for tensor-train compression of ConvL-STM [28]. **(1)** Since the best ConvLSTM baseline has filter size $K = 5$, we only consider $K = 5$ in the compression experiments. **(2)** We observe that the compressed ConvLSTM models consistently achieve better performance with learning rate $10^{-3}$. **(3)(4)** The compressed ConvLSTMs are robust to different order and ranks, and $N = 2, R = 8$ wins by a small margin.

| Filter size $K$ | Learning rate | Order of TTD $N$ | Ranks of TTD $R$ |
|---|---|---|---|
| 5 | $\{10^{-4}, 10^{-3}\}$ | $\{2, 3\}$ | $\{8, 16, 32\}$ |

Table 7: Hyper-parameters search values for Tensor-Train compression of ConvLSTM.

## B.5 Datasets

**Moving-MNIST-2 dataset.** We generate the Moving-MNIST-2 dataset by moving two digits with size $28 \times 28$ in the MNIST dataset within a $64 \times 64$ black canvas. These digits are placed at a random initial location and move with constant velocity in the canvas and bounce when they reach the boundary. Following Wang et al. [6], we generate 10,000 videos for training, 3,000 for validation, and 5,000 for test with default parameters in the generator[3].

**KTH action dataset.** The KTH action dataset [20] contains videos of 25 individuals performing six types of actions on a simple background. Our experimental setup follows Wang et al. [6], which uses persons 1-16 for training and 17-25 for testing, and we resize each frame to $128 \times 128$ pixels. We train all our models to predict 10 frames given 10 input frames. We randomly select 20 contiguous frames from the training videos as a sample and group every 10,000 samples into one epoch to apply the learning strategy, as explained at the beginning of this section.

**Something-Something V2 dataset.** The Something-Something V2 dataset [21] is a benchmark for activity recognition, which can be download online[4]. Following Wang et al. [7], we use the official subset with 41 categories that contains 55111 training videos and 7518 test videos. The video length ranges between 2 and 6 seconds with 24 frames per second (fps). We reserve $10\%$ of the training videos for validation, and use the remaining $90\%$ for optimizing the models.

## B.6 Evaluation Metrics

We use two traditional metrics, MSE (or PSNR) and SSIM [51], and a recently proposed deep-learning-based metric LPIPS [52], which measures the similarity between features from different

layer. Since MSE (or PSNR) is based on pixel-wise difference, it favors vague and blurry predictions — thus, it is not a proper measurement of perceptual similarity. While SSIM was initially proposed to address the problem, Zhang et al. [52] shows that their proposed LPIPS metric aligns better with human perception.

### B.7 Ablation Studies

Here, we show that our proposed Conv-TT-LSTM consistently improves the performance of ConvLSTM, regardless of the architecture, loss function, and learning schedule used. Specifically, we perform three ablation studies on our experimental setting, by **(1)** Reducing the number of layers from 12 layers to 4 layers (same as [4] and [6]); **(2)** Changing the loss function from $\mathcal{L}_1 + \mathcal{L}_2$ to $\mathcal{L}_1$ only; and **(3)** Disabling the scheduled sampling and use teacher forcing during training process. We compare the performance of our proposed Conv-TT-LSTM against the ConvLSTM baseline in these ablated settings, Table 8. The results show that our proposed Conv-TT-LSTM consistently outperforms ConvLSTM in all settings, i.e., the Conv-TT-LSTM model improves upon ConvLSTM in a board range of setups, which is not limited to the specific setting used in our paper. These ablation studies further show that our setup is optimal for predictive learning in Moving-MNIST-2 dataset.

| Model | | Layers | | Sched. | | Loss | | (10 -> 30) | | | Params. |
|---|---|---|---|---|---|---|---|---|---|---|---|
| | | 4 | 12 | TF | SS | $\ell_1$ | $\ell_1 + \ell_2$ | MSE | SSIM | LPIPS | |
| ConvLSTM | - | ✓ | × | × | ✓ | × | ✓ | 37.19 | 0.791 | 184.2 | 11.48M |
| Conv-TT-LSTM | FW | ✓ | × | × | ✓ | × | ✓ | **31.46** | **0.819** | **112.5** | **5.65M** |
| ConvLSTM | - | × | ✓ | ✓ | × | × | ✓ | 33.96 | 0.805 | 184.4 | 3.97M |
| Conv-TT-LSTM | FW | × | ✓ | ✓ | × | × | ✓ | **30.27** | **0.827** | **118.2** | **2.65M** |
| ConvLSTM | - | × | ✓ | × | ✓ | ✓ | × | 36.95 | 0.802 | 135.1 | 3.97M |
| Conv-TT-LSTM | FW | × | ✓ | × | ✓ | ✓ | × | **34.84** | **0.807** | **128.4** | **2.65M** |
| ConvLSTM | - | × | ✓ | × | ✓ | × | ✓ | 33.08 | 0.806 | 140.1 | 3.97M |
| Conv-TT-LSTM | FW | × | ✓ | × | ✓ | × | ✓ | **28.88** | **0.831** | **104.1** | **2.65M** |
| Conv-TT-LSTM | SW | × | ✓ | × | ✓ | × | ✓ | 25.81 | **0.840** | **90.38** | 2.69M |

Table 8: Evaluation of ConvLSTM and our Conv-TT-LSTM under ablated settings. In this table, FW stands for *fixed window approach*, SW stands for *sliding window approach*; For learning scheduling, TF denotes *teaching forcing* and SS denotes *scheduled sampling*. The experiments show that **(1)** our Conv-TT-LSTM is able to improve upon ConvLSTM under all settings; **(2)** Our current learning approach is optimal in the search space; **(3)** The sliding window approach outperforms the fixed window one under the optimal experimental setting.

## C  Additional Experimental Results

**Per-frame evaluations.** The per-frame metrics are illustrated in Figure 7 for Moving-MNIST-2 dataset, and Figure 8 for KTH action dataset. **(1)** In the Moving-MNIST-2 dataset, PredRNN++ performs comparably with our Conv-TT-LSTM on early frames, but drops significantly for long-term prediction. E3D-LSTM performs similarly to ConvLSTM baseline, and our Conv-TT-LSTM consistently outperforms E3D-LSTM and ConvLSTM over all frames. **(2)** In the KTH action dataset, PredRNN++ consistently perform worse than our Conv-TT-LSTM model for all frames; E3D-LSTM performs well on early frames in MSE and SSIM, but quickly deteriorates for long-term prediction.

**Additional visual results: Video prediction.** Figure 9, 10, 11, 12, 13, and 14 show additional visual comparisons. We also attach two video clips (KTH and MNIST) as supplementary material.

**Additional visual results: Early activity recognition.** We attach two video clips (video 1 and 2) as supplementary material. The videos show the comparisons among 3D-CNN, Conv-LSTM and our Conv-TT-LSTM when the input frames are partially seen. The time-frame of the video corresponds to an amount of video frames seen by the models.

Figure 7: **Frame-wise comparison in MSE, SSIM and PIPS on Moving-MNIST-2 dataset.** For MSE and LPIPS, lower curves denote higher quality; while for SSIM, higher curves imply better quality. Our Conv-TT-LSTM performs better than ConvLSTM baseline, PredRNN++ [6] and E3D-LSTM [7] in all metrics (except for PredRNN++ in term of MSE).

Figure 8: **Frame-wise comparison in PSNR, SSIM and PIPS on KTH action dataset**. For LPIPS, lower curves denote higher quality; For PSNR and SSIM, higher curves imply better quality. Our Conv-TT-LSTM outperforms ConvLSTM, PredRNN++ [6] and E3D-LSTM [7] in SSIM and LPIPS.

Figure 9: 20 frames prediction on KTH given 10 input frames. Every 2 frames are shown.

Figure 10: 20 frames prediction on KTH given 10 input frames. Every 2 frames are shown.

Figure 11: 20 frames prediction on KTH given 10 input frames. Every 2 frames are shown.

Figure 12: 20 frames prediction on KTH given 10 input frames. Every 2 frames are shown.

Figure 13: 30 frames prediction on Moving-MNIST given 10 input frames. Every 3 frames are shown. The first frames ($t = 1$ and 11) are animations. To view the animation, Adobe reader is required.

Figure 14: 30 frames prediction on Moving-MNIST given 10 input frames. Every 3 frames are shown. The first frames ($t = 1$ and 11) are animations. To view the animation, Adobe reader is required.

## Footnotes

[4]`https://20bn.com/datasets/something-something`