[Reviews · NeurIPS 2020]

Review 1

Summary and Contributions: After reading the rebuttal I have changed my score to 7 with the following feedback: 1: Certainly the comparison with Yang et. al. and the mention that the two approaches use Tensor Train in completely different manners is important. 2: I concur with R2 that it is also important to show performance devoid of all scheduling tricks and on a 'standardized' depth network commonly used in other papers for benchmarking purposes. This would remove any doubts in the mind of the reader with regards to these techniques somehow synergistically favoring HO TT RNN over previous methods. Overall good paper. The paper proposes a variant of Higher-order convolutional recurrent neural network for the task of supervised learning on video sequences. The key idea behind higher order CRNN is to capture the smooth evolution of frames in videos. As there is strong connection between neighboring frame data, it is important to capture this with higher order RNNs which correlate information across several time steps (as opposed to pure RNNs which correlate across typically one timestep only). To enable efficient computation of the above higher order CRNN, the paper proposes a tensor decomposition of the higher order recurrent convolution operation which correlates information across the past several frames. Naively implemented, this operation would take O(N^3) time. The paper presents a key link between the evolution of data across frames in video (i.e. smoothly evolving over time in an autoregressive manner) and tensor-train decomposition (decomposing a large tensor into a chain of tensor products). Validation on the above proposed architecture is presented, which outperforms previous work while maintaining fewer number of learnable parameters.

Strengths: The paper is well presented with well thought out rationale and reasoning. I particularly like the implicit link between video data and tensor train decomposition. The tensor train decomposition with the accompanying linear time computation algorithm for higher order computation is well presented, and improves the computational complexity of evaluating the higher order convolutional RNN. The empirical validation is comprehensive and quite convincing of the merits of this approach vs with respect to CLSTM and PredRNN++. The qualitative results also seem impressive and I'm fairly convinced that the proposed method does a better job at solving video based tasks than the compared work.

Weaknesses: The two major weaknesses are a lack of comparison to previous work by Yang et. al. [1] that also utilizes tensor train decomposition, as well as possibly incomplete validation with regards to efficiency. The work by Yang et. al. is somewhat similar in its application of tensor train to reduce the complexity of computation. Although, it does not rely on the same structure as this work (smooth evolution over time in video data vs tensor train), it does rely on somewhat of a similar structure (i.e. decomposing a complex tensor operation on image data as a tensor train operation akin to convolution). Looking at these two side by side, I appreciate their difference, however I think they're still too similar to not require a comparison. One could conceivably imagine that the same underlying structure is exploited by both approaches, which diminishes the novelty of the work. It remains to be seen whether this application of tensor train is orthogonal to the application of tensor train by Yang et. al. It is indeed important to verify whether the proposed application of tensor train is sufficiently different to previous work so as to confirm that this work is not aliasing previous work by Yang et. al. Without this comparison, it leaves the doubt that the underlying mechanism of improvement might be the same as previous work. Another issue with the current work is with respect to efficiency evaluation. As this is a central point in the paper, it should be fairly and accurately presented. I didn't see any specific runtime or latency evaluation with regards to the proposed architecture. The reason why this is an issue is because HO-RNN, specifically this version of HO-RNN require a chained dependency to evaluate the higher order term in order to reduce the amount of FLOPs required. Although this tradeoff between FLOPs, and latency is likely useful, it should be presented in the work. It should be noted that although the naive O(N^3) algorithm requires more computation, it can be accelerated on SIMD architecture such as GPUs due to a lack of dependency structure. This aspect of the work requires more discussion and presentation, and the paper is somewhat incomplete without it. Although I am somewhat skeptical at this time, I can be convinced to change my views. I think this work is solid, and I thoroughly appreciate the unique link presented between tensor train and HO-CRNN. I think this is worth highlighting as a contribution, and validating with showing exactly how the tensor train captures this smooth evolution of video frames over time. It behooves the author to do this due diligence to highlight the importance and novelty of the work.

Correctness: I see no major issue in this regard

Clarity: The paper is mostly well written. However, I did find the abundance of notation to be somewhat hampering in understanding the work. Perhaps the authors can think of a way to simplify the presentation, or reduce the amount of notation.

Relation to Prior Work: The previous work is mostly well discussed, with exceptions. I have discussed this issue in the weaknesses section.

Reproducibility: Yes

Additional Feedback:


Review 2

Summary and Contributions: This manuscript proposes a new LSTM architecture that combines ConvLSTM and HO-LSTM and decomposes the weight tensors with TT while re-using core tensors. The model is supposed to capture high order spatial temporal pattern in video data. And experimental results show improved modeling performance and reduced computation cost.

Strengths: It seems an interesting idea to exploit the topology of tensor-train decomposition, by assigning the core tensors to consecutive time steps, and using only a subset of them to construct a tensor as part of the \Phi function in HO-LSTM. The authors also did a great job in providing sufficient experimental results, including the appendices.

Weaknesses: Motivation: I understand that it is sometimes challenging for RNNs to model high order interaction between past states in an explicit manner. However, I would not argue, as the authors do in the introduction, that it is due to the 1st order Markovian structure. RNNs are supposed to memorize last hidden states due to the recursive definition. [Siegelmann and Sontag 1995] even claims RNNs to be Turing complete. In practice, it is often the vanishing/exploding gradients that prevent RNNs from learning such high order interaction. Usecase: I also doubt to which extent is high order interaction really necessary in predicting activity classes and next frames. The performance of optical flow features in these tasks seems to suggest one only needs the latest frames. [Yu et al. 2017], on the contrary, provided a more convincing use case where high order interaction might improve the modeling of sensor data. Implementation details: From line 158-168, I have the impression that such an architecture has to be trained with so many scheduling tricks, that it might not provide a robust solution to general tasks. I would suggest performing ablation studies to factorize the contribution in a quantitative fashion. Furthermore, the choice of a 12 layered RNN for MNIST and KTH data seems an overkill to me and it raises the question, i) whether it is the increased model capacity or the proposed CTTD module that is responsible for the performance and ii) do the baseline RNNs, i.e. ConvLSTMs, also have 12 layers… Misc: Regarding the pre-processing: I wonder if it conflicts with the paper’s ambition to model high order interaction. I would not include videos in a proper publication. Please consider including them in the appendices. In Line 86, I believe the (t) after X should not be in superscript? ###################################### Update: I do acknowledge the new way to integrate TT into the high order RNN and have to admit that I missed a couple of detailed things in the appendix. I do like the new table and thank you very much for raising the point! However, I still have the concern about the 4-layered and 12-layered RNN architectures and the training scheduling. I would expect a paper which proposes a new method to focus on and highlight the core contribution instead of -in order to outperform all possible benchmarks- introducing all possible tricks. The ablation study in the appendix is not complete in that I do not see a comparison between with- and without scheduling while all other configuration remains the same (second row in tab. 4 appendix). Furthermore, since new method "consistently outperforms ConvLSTM in all settings", it would be -in my personal opinion- better to report the performance in a simpler and more transparent setting such as the new table in the rebuttal. To this end, I can not yet tell, as I wrote in my first review, whether it is the new method or the 12 layers and scheduling that contributes to the improved performance. I would have given the paper a much higher score if the experiments were more transparent, even if the results were a little bit worse. I do appreciate the GREAT GREAT effort that the authors have put into the work. Having also taken into account the feedback as well as the opinions of other reviewers, I will update my score to be "marginally below" but will not protest if the paper gets accepted.

Correctness: Seems correct to me.

Clarity: The paper, especially the methodology part, was not easy to follow partially due to the over complicated notation system. However, the appendix turns out to be helpful.

Relation to Prior Work: Yes. Most of the relevant works, to the best of my knowledge, have been covered properly.

Reproducibility: Yes

Additional Feedback:


Review 3

Summary and Contributions: This paper proposed a Tensor-Train-like convolutional structure, convolutional tensor-train decomposition(CTTD), and constructed a CTTD based higher-order convolutional LSTM model. This model can efficiently learn long-term Spatio-temporal correlations, along with a succinct representation of history. The low-rank design jointly reduces the number of parameters and naturally encodes temporal dependencies. The experiments show that the model outperforms existing approaches while using only a fraction of parameters.

Strengths: 1. The model is presented clearly and the figures are easily understood. The experimental results were presented to demonstrate the effectiveness of the proposed idea. 2. A higher-order convolutional structure can naturally learn information of long-term temporal input. But the number of parameters will be too large which leading a difficulty into the training process. The proposed tensor-train structure can reduce the number of parameters easily.

Weaknesses: 1. Some explanations are needed. In Line 151, I concern about why "implicit regularizer" leading to "generalized models". There is no analysis on it. 2. In Table 4. It would be better to state the difference between Conv-LSTM[4] and Conv-LSTM(baseline) clearly.

Correctness: The claims and methods are correct.

Clarity: The paper is generally well written and also well organized. In Figure 1, "G" -> \mathcal{G}.

Relation to Prior Work: Yes.

Reproducibility: Yes

Additional Feedback: ================= Post-rebuttal ==================== About the experiments, although the result is good, it is difficult to distinguish whether the model itself is the main contributor to the performance. Therefore, it is better to make the experiments more transparent as R2 said. And I mainly concerned that using the information bottleneck to interpret the model. Specifically, in the author's feedback, the authors said low-dimension could lead to less redundant. However, useful information may be also reduced at the same time. And there is not any evidence in the paper show the changes of information. Thus, this claim is not correct. In conclusion, considering the interesting structure and good empirical result, I would maintain the score 6. If this paper is accepted, it is not suggested to use the information bottleneck to interpret the CTTD. Alternatively, simply using the reducing noise attribution of decomposition is enough rather than an uncertain theory.

[Author Response · NeurIPS 2020]

**Paper ID 10989: Convolutional Tensor-Train LSTM for Spatio-Temporal Learning.** We thank all reviewers for
their valuable feedback. Reviewers found that this work is solid, the main idea is interesting and clearly presented.
Below is the point-by-point responses to the reviewers.

**R1-Q1. Comparison to Yang et. al. [40]:** R1 appreciates the difference between ours and [40], especially the unique
link between tensor train and HO-CRNN. There are two major differences between our work and [40]: **(1)** while [40]
relies on the classic tensor-train decomposition (TTclassic) based on [18], our *convolutional tensor-train decomposition*
(CTTD, Eq.(6)) factorizes the tensor with convolutions instead of inner products; **(2)** [40] compresses only input-hidden
weights within one time-step in LSTM. Our CTTD compresses spatio-temporal interactions over time. This is a
new design of tensor-train decomposition for spatio-temporal data. We add a new comparison between ours and
[40] in the table below. For fair comparison, we applied the core idea of [40] to ConvLSTM (baseline) as follows:
$[\mathcal{I}(t); \mathcal{F}(t); \tilde{\mathcal{C}}(t); \mathcal{O}(t)] = \sigma(\text{TTclassic}(\{\mathcal{W}_i\}) * \mathcal{X}(t) + \mathcal{K} * \mathcal{H}(t-1))$. From the comparison, we observe that our
model outperforms [40] on MNIST and KTH (except LPIPS on KTH) with similar number of parameters.

Furthermore, we believe our work is orthogonal to [40] —
[40] can be used to further compress our model by decom-
posing each factor $\mathcal{G}(i)$ (in Eq.(6)) with TTclassic.

| Dataset | TTclassic [40] | | ConvLSTM | | Conv-TT-LSTM | |
|---|---|---|---|---|---|---|
| (predicted frames) | SSIM | LPIPS | SSIM | LPIPS | SSIM | LPIPS |
| MNIST (10) | 0.890 | 59.09 | 0.882 | 67.13 | **0.915** | **40.54** |
| MNIST (30) | 0.817 | 125.2 | 0.806 | 140.1 | **0.840** | **90.38** |
| KTH (20) | 0.900 | **120.1** | 0.903 | 137.1 | **0.907** | 133.4 |
| KTH (40) | 0.874 | **163.5** | 0.876 | 201.3 | **0.882** | 191.2 |
| Params. | 2.20M | | 3.97M | | 2.69M | |

**R1-Q2. Efficiency evaluation.** We agree that there is a
trade-off between FLOPs and latency. Therefore, we intro-
duce two algorithms in Appendix A. While Alg. 2 signifi-
cantly decreases the complexity in FLOPs, it also lowers the degree of parallelism. However, Alg. 1 shows how our
model can be parallelized. Ideally, these two algorithms can be combined using CUDA multi-streams (execute multiple
kernels in parallel): use Alg. 1 for the beginning iterations of $i$ and Alg. 2 for the later ones (the beginning ones have
smaller kernel sizes). In our current implementation, we use Alg. 2 to reduce the GPU memory requirement. The
run-time of current implementation is 27.3 mins (37.83 GFLOPs) for Conv-TT-LSTM and 26.2 mins (55.83 GFLOPs)
for ConvLSTM (per epoch on KTH). We will add the run-time and this discussion in the final version.

**R2-Q1. Motivation of higher-order RNN.** While vanilla RNN can be a universal approximator or Turing complete
theoretically, there is no guarantee that the model will find the optimal solution. R2 believes that in practice, the
vanishing/exploding gradients prevent RNNs from learning higher-order interaction. Our proposed model addresses the
vanishing/exploding gradient problem by incorporating long-term dependencies with higher-order RNNs [8].

**R2-Q2. Use case of higher-order RNNs.** The applications in our experiments requires future prediction. To predict
the most possible future, understanding the long-term dynamics is essential. The early activity recognition task requires
a model to understand the dynamics of the video, so the model can predict an activity at the early stage. For the video
prediction tasks, the model learns to predict 10 frames during training. In testing, the model further predicts 30-40
frames. Optical flow features (based on short-term dynamics) are not sufficient to make such predictions.

**R2-Q3. Add ablation studies. The choice of a 12 layer baseline.** The paper already includes the detailed ablation
studies in Table 3 and Appendix B.6: single v.s. higher-order model, the necessity of convolutions in CTTD, 4 v.s. 12
layers, the benefit of scheduling tricks, etc. The results show that Conv-TT-LSTM consistently outperforms ConvLSTM
baseline under all scenarios. Necessity of a deep model for video prediction has been already discussed in [43].

**R2-Q4. Many scheduling tricks.** These scheduling tricks are used for both our model and the baseline ConvLSTM, so
our performance improvement is NOT due to the scheduling tricks (also shown by the aforementioned ablation studies).
These scheduling tricks are commonly used for prediction and early activity recognition tasks [19,23-24].

**R2-Q5. Pre-processing conflicts with the paper's motivation (higher-order interaction).** The reviewer misunder-
stood the difference between *order in RNNs* and *order of a tensor decomposition*. The former refers to the number of
previous time-steps used at each update, while the latter denotes the number of factors in tensor decomposition. In fact,
we propose the pre-processing module mainly to decouple these two concepts (explained in Lines 136-146), and the
order of tensor decomposition only controls the complexity of the mapping function $\Phi$.

**R2-Q6. Include videos in the appendix.** '10989_result_videos' in the supplementary file already includes the videos.

**R3-Q1. Analysis of "implicit regularizer" leading to "generalized models".** The relationship between low-rank
regularizer and generalized models is discussed in Line 40-42, and analyzed theoretically in [11, 12]. Intuitively, consider
a linear model $\mathbf{y} = \mathbf{A}\mathbf{x}$, where $\mathbf{A} \in \mathbb{R}^{q \times p}$ is factorized as $\mathbf{A} = \mathbf{U}\mathbf{V}$ with $\mathbf{U} \in \mathbb{R}^{q \times r}, \mathbf{V} \in \mathbb{R}^{r \times p}, r < \min(p, q)$. The
factorization implicitly creates a bottleneck $\mathbf{h}$ since the model can be evaluated as $\mathbf{h} = \mathbf{V}\mathbf{x}$ and $\mathbf{y} = \mathbf{U}\mathbf{h}$. Since $\mathbf{h}$ has
lower dimension than $\mathbf{x}$, the bottleneck filters out redundant information, leading to more generalized models.

**R3-Q2. Difference between ConvLSTM [4] and ConvLSTM (baseline).** ConvLSTM [4] is the original model
proposed by [4]. We re-implemented ConvLSTM of [43] as our baseline. It has exactly the same model size, skip
connections and uses the same training strategies as our Conv-TT-LSTM. We will add the clarification in the paper.

[Meta-Review · NeurIPS 2020]

This paper develops a higher-Markov-order convolutional LSTM based on tensor train decomposition, with applications to spatio-temporal activity analysis in videos. The reviews were mixed but marginally positive on average, and the scores increased slightly following the rebuttal and some discussion.There is a consensus that the approach is novel and interesting. The main criticism is that despite the extensive experiments it remains unclear whether it is novel formulation itself that is producing the observed improvements, or the many other points that differ relative to the baselines. The advantages of using Markov order>1 in this application also need to be clarified. Overall, the AC and SAC agreed that this was above threshold for NeurIPS. However the final version needs to do its best to address the concerns of the reviewers.